# Activation of Genes by Nuclear Receptor/Specificity Protein (Sp) Interactions in Cancer

**DOI:** 10.3390/cancers17020284

**Published:** 2025-01-17

**Authors:** Stephen Safe, Evan Farkas, Amanuel E. Hailemariam, Arafat Rahman Oany, Gargi Sivaram, Wai Ning Tiffany Tsui

**Affiliations:** Department of Veterinary Physiology and Pharmacology, College of Veterinary Medicine, Texas A&M University, College Station, TX 77843, USA; farkasea@tamu.edu (E.F.); ahailemariam@tamu.edu (A.E.H.); arafatr@tamu.edu (A.R.O.); gs715@tamu.edu (G.S.); wntsui@exchange.tamu.edu (W.N.T.T.)

**Keywords:** nuclear receptor, Sp transcription factors, NR/Sp complexes, gene regulation

## Abstract

Nuclear receptors (NRs) and specificity protein (Sp) transcription factors (TFs) play major roles in carcinogenesis through their regulation of procarcinogenic pathways and genes. This review summarizes an important sub-set of genes coregulated by NR/Sp in cancer cells and is focused on the mechanisms of these responses that involve NR acting as a ligand-dependent cofactor of DNA-bound Sp TFs. In addition, several NR-regulated genes involve both NRs and Sp TFs bound to GC-rich Sp binding sites that can be proximal or distal and this includes interactions with other nuclear cofactors and the basal transcriptional machinery.

## 1. Introduction

Specificity protein (Sp) transcription factors (TFs) include Sp1, Sp2, Sp3 and Sp4. They exhibit structural and functional similarities and are members of the Sp/Kruppel-like factor (Sp/KLF) family [1,2] (Figure 1A). Sp1 was the first TF identified, and the Sp1–4 subfamily is particularly important in cancer. The Sp1, Sp 3 and Sp4 TFs interact with cis GC-rich promoter sequences to activate gene expression. However, Sp3 has an inhibitory domain that can also result in inhibition of the expression of some genes in a cell context-dependent manner. The role of Sp TFs in tumorigenesis has been outlined in recent reviews [1,2,3,4,5,6], where it was pointed out that the transformation of human fibroblasts, mammary epithelial cells and human smooth muscles into tumor cells, is accompanied by the increased expression of Sp1 and other Sps [7,8,9,10]. There is also extensive evidence from Sp knockdown studies that Sp-deficient cancer cell lines exhibit decreased cell growth, survival migration and invasion [6,11]. These effects were observed in Sp1-, Sp3- and Sp4-deficient Panc1 pancreatic cancer cells. Analysis of the differentially expressed (DE) genes after Sp knockdown showed that the number of commonly regulated genes after a combined knockdown of Sp1/Sp4, Sp1/Sp3 and Sp3/Sp4 was 1140, 1113 and 2753, respectively. For Sp3 and Sp4, 50% of all Sp3 and 64% of all Sp4-regulated genes were regulated in common. However, despite the large number of genes commonly regulated by Sp1, Sp3 and Sp4, the functional effects of individual Sp1, Sp3 and Sp4 knockdown in Panc1 cells were comparable and the loss of one Sp protein was not compensated for by the other two TFs [11]. Moreover, since Sp1, Sp3 and Sp4 regulate multiple genes required for cancer cell proliferation, survival and invasion, it has been suggested the Sp TFs are non-oncogene addiction genes [11].

The nuclear receptor (NR) superfamily of transcription factors regulates genes and pathways that play critical roles in maintaining cellular homeostasis and in pathophysiology where NRs are major drug targets [12]. Humans express 48 NRs that exhibit a common domain structure (Figure 1B), and they are classified into seven different groups based on homology (Figure 1C). For example, group III includes both estrogen receptors (ERα, ESR1 and ERβ, ESR2), the androgen receptor (AR, NRC34), progesterone receptor (PR, NR3C3) and estrogen receptor-related receptor (ERRα, NR3B1 and ERRβ, NR3B2). With the exception of dosage-sensitive sex reversal–adrenal hypoplasia critical region on chromosome X gene1 (DAX1) and short heterodimeric partner (SHP), which do not contain a DNA-binding domain (DBD), members of the other six groups are all ligand-activated receptors that bind their cognate response elements in target gene promoters. The cis-elements that interact with NRs show some variability with respect to their sequences, and NRs can bind as monomers, dimers and heterodimers, in which the retinoid X receptor is a partner. For example, the orphan nuclear receptor 4A1 (NR4A1, Nur77) binds the Nur77 binding response element (NBRE, AGGTCA) as a monomer, a Nur response element (NuRE, AAATG/AC/TCA) as a dimer and also a DR5 sequences (AAGTCA-N_5_-AGGTCA) as a heterodimeric NR4A1:RXR complex [13,14,15]. Some NRs and their ligands also regulate gene expression by not binding directly to promoter DNA but through interactions with other DNA-bound transcription factors [16,17]. It should also be noted that NR ligands can be defined as agonists, antagonists and inverse agonists, which involve ligand-dependent activation, the inhibition of agonist-induced activation and the repression of NR-dependent transactivation, respectively. The activity of NR ligands as agonists, antagonists and inverse agonists can be observed for NRs bound directly to target gene promoters or bound to Sp TFs.

Several pro-oncogenic genes regulated by Sp1, Sp3 and Sp4 utilize NRs as ligand-dependent cofactors, and these will be discussed in this review. The NR/Sp mechanism of gene regulation can be modified by receptor ligands, and this provides an additional pathway for targeting Sp-regulated genes. This review highlights published articles on NR/Sp-mediated genes, and it is likely that many other genes, yet to be identified, are regulated by this pathway, since NRs have been understudied as nuclear cofactors.

## 2. Subgroup III Receptors and Their Interactions with Sp TFs

Subgroup III receptors include the well-characterized steroid hormone receptors, which play a critical role in maintaining cellular homeostasis but also in pathophysiology, where they are major drug targets. This group will be discussed initially since their interactions with Sp TFs have been extensively characterized.

Estrogen receptor: The hormonal regulation of ER-responsive genes has been investigated in breast and other cancer cell lines, and induction or repression of many of these genes by ER ligands primarily involve ER interactions with DNA-bound Sp1 and, in a few cases, Sp3 or Sp4. The results in Figure 2 summarize studies showing that ERα interacts with Sp1 to activate several genes in cancer cell lines, and these include TNFα [18], E2F1 [19,20], BRCA1 [21], insulin-like growth factor receptor 1 (IGFR1) [22], LDL receptor [23], multidrug resistance (MDR) gene [24], cyclin D1 [25], bcl2 [26], adenosine deaminase [27] and retinoic acid receptor α (RARα) [28], cfos [29], IGFBP4 [30], thymidylate synthase [31], DNA polymerase α [32], carbamyl phosphate synthetase/aspartate carbonyl-transferase/dihydroorotase (CAD) [33], Hsp27 [34], VEGF [35] and MCL-1 [36]. In addition, hormone-dependent downregulation of VEGF [37] and vascular endothelial growth factor receptor 2 (VEGFR2, KDR) [38] and protein S (PROS1) gene [39] expression, which involved ERα/Sp3 and recruitment of corepressors including RIP140 and the NCoR-SMRT-HDAC3 to the PROS1 gene promoters (Figure 3). ERβ interacts with Sp1 bound to the ERα gene promoter, resulting in the recruitment of nuclear corepressors and decreased expression of ERα [40] and elevated levels of ERβ competitively displaced ERα/Sp1-mediated binding to the BRCA2 gene promoter to decrease transactivation [41]. In addition, ERα/Sp1 also activates receptors for advanced glycation end products (RAGE) [42], cyclooxygenase 1 (COX1) [43] and rat SK3 [44] genes in non-transformed cell lines.

A summary of the ER/Sp1 regulated genes in cancer (Figure 2) clearly demonstrates that induced (or repressed) gene products are major factors in hormone-induced carcinogenesis. The interacting domains of ERα and Sp1 were investigated in a series of ERα and ERβ domain-swapping experiments and deletion studies; multiple regions of ERα were required for E2-induced transactivation, and these differed from those required for antiestrogens [45,46]. Exon-deleted mutants of ERα were investigated with respect to ERα/Sp1-mediated induction of VEGF and the exon3 (ER∆3) but not exon5 or exon7 mutants induced transactivation [47]. Fluorescence resonance energy transfer (FRET) experiments also demonstrated a critical role for the C-terminal DNA binding domain (DBD) of Sp1 for interacting with ERα [48]. It was also observed that vitamin D receptor-interacting protein 150 (DRIP150) coactivated ERα/Sp1 in breast cancer cells [49].

Several studies showed that hormone-induced ER/Sp1 mediated gene expression not only required Sp1 binding sites but also cis-ER-responsive element (ERE) half-sites (ERE^1^/_2_) that bind ERα. Krishnan and coworkers first characterized an ER-Sp-like sequence in the cathepsin D gene promoter (GGGCGG(N)_23_-ACGGG) and showed that transactivation of cathepsin D promoter constructs required intact Sp1 and ERE^1^/_2_ sites (Figure 4). Although the role of ERα as a monomer or dimer has not been determined, interactions of ER and Sp1 (ER-Sp1) bound to proximal and distal sequences can be functional [50,51]. In subsequent studies, cis-acting GC-(N)_x_-ERE^1^/_2_ sites have been identified in the transforming growth factor α (TGFα) [52] progesterone receptor [53], cyclin G2 [54], MDR1 [55], Hsp27 [56], rat creatine kinase B genes [57] and metastasis-associated protein 3 (MTA3) [58]. Thus, both ERE^1^/_2_ and GC-rich sites are required for maximal hormone-induced activation of ER-Sp1, and ERβ competitively inhibits ER-Sp1-mediated induction of Hsp27 in papillary thyroid cancer cells [56]. Mouse claudin-5 is also an ER-Sp1-regulated gene in mouse brain microvascular epithelial cells [59]. Most of the ER/Sp1-regulated genes (Figure 2) are induced by E2; however, the effect of ER-dependent activation/inactivation of Sp regulated genes is complex and depends not only on the cis promoter elements but also on ligand structure [46,60,61]. Kim and coworkers [46] showed that ligand-dependent activation of luciferase construct containing 3 tandem Sp1 binding sites (pSp1_3_-luc) required different ER domains for induction of luciferase by E2. Moreover, the antiestrogen ICI182780 also induced luciferase in breast cancer cells transfected with pSp1_3_-luc but different domains of ERα were required for this response. In contrast, the activity of ERβ/Sp1 was minimal. Ligand-dependent induction of luciferase activity in breast cancer cells transfected with pSp1_3_-luc was also induced by E2 and structurally-diverse estrogenic endocrine disruptors and responses varied with respect to ERα (wild = type/mutations) and expression of Sp1, Sp3 and Sp4 [60]. Results of Sp-knockdown studies suggests that while Sp1 is dominant, the activation of ER/Sp by some ligands may also include Sp3 or Sp4 [61].

Progesterone receptor (PR): The PR also interacts directly with Sp1, and PR ligands activate both PR/Sp1 and PR-Sp1 (PRE-(N)_X_-GC). Genes that are induced by PR/Sp1 have primarily been observed in T47D breast cancer cells that overexpress PR and include p21/p27 [62,63], EGFR [59], PRL receptor [64] and MKP1 [65] that are activated by PR/Sp1; HSPB8 is induced via PRE-(N)_X_-GC sequence and also includes cyclin D1 as part of the complex [66]. In addition, PR/Sp1 is involved in activating endothelial nitic oxide synthase expression in endothelial cells [67]. Thus, PR interaction with Sp1 modulates only a limited number of genes compared to ERα-Sp interactions in cancer cells.

Androgen receptor-Sp interaction: AR interacts with Sp1 bound to GC-rich sites in the AR promoter, and the flavonoid quercetin inhibits this response, which could be partially reversed with JNK inhibitors. It was concluded that the inhibitory effects of quercetin were associated with an AR/Sp1 complex that also includes c-Jun [68], and another study confirmed the importance of the Sp1 binding sites [69]. A recent report showed that the DHT treatment of prostate cancer cells induced the formation of nuclear AR/Sp1 and AR/Sp3 complexes on the GPER1 gene promoter to decrease transactivation [70]. In contrast, AR/Sp1 formation on the cell surface type-A receptor 3 (EPHA3) promoter contributed to the expression of this gene which was induced by DHT in prostate cancer cell lines [71]. The androgen-responsiveness of the p21 promoter in prostate cancer cells is also due to AR/Sp1 interaction [72], whereas the induction of the AR coactivator, nuclear receptor interaction protein (NRIP), involves AR-Sp, which includes an androgen response element (ARE) and GC-rich sites [73]. AR/Sp1 is also involved in the regulation of luteinizing hormone β in pituitary cells, and this response involves other factors [74].

Other group III receptor interactions with Sp1: The glucocorticoid receptor interacts with Sp1 in the regulation of monoamine oxidase A [75] and B [76] in neuronal-derived cancer cells, and estrogen receptor-related receptor (ERR) interacts with Sp1 to activate p21 in Hela cells [77]. Other examples of GR/Sp1 (NR3C1/Sp1) mineralocorticoid receptor/Sp1 and ERR/Sp were also observed in a non-cancer cell context [78,79,80,81,82].

## 3. Subgroup II Receptor and Their Interactions with Sp TFs

Endogenous ligands for HNF4 (NR2A), RXR (NR2B), TR (NR4C), TLX (NR2E1), PNR (NR2E2), COUP-TF (NR2F) and EAR2 (NR2F6) have not been identified; RXR binds to 9-cis-retinoic acid, but this is not a confirmed biological ligand. Some fatty acids and synthetic chemicals have been identified as ligands for some of these receptors. The interaction of subgroup II receptors with Sp TFs in cancer cells is limited.

Retinoid X receptor (RXR): Since RXR is a binding partner for multiple NRs and interaction of RXR, receptor heterodimers with Sp1 may involve RXR and also the receptor partner individually or in combination. In cancer cells, RARα/RXRα (NR1B1/NR2B1) interacts with Sp1 and Sp3 on the 17β-hydroxysteroid dehydrogenase type 2 (HSD17B2) promoter, and retinoic acid (RA) induces its expression [83]. However, RA induces thrombomodulin gene expression in pancreatic cancer cells, and this involves distal interactions of heterodimeric receptor complex with two genomic DR4 binding sites (−1531 to −1516) and Sp1 (−145 to −121) sites [84]. Several other studies reported that RXR interactions with Sp1 were involved in the activation of the ATP-binding cassette transporter 1 (ABCA1; LXR partner) [85], urokinase (RAR partner) [86,87], tissue-type plasminogen activator (t-PA; RAR partner) [88] and acyl-coenzyme A oxidase (ACO; PPAR partner) [89] in non-cancer cell lines.

Hepatocyte Nuclear Factor-4 (HNF-4): Interactions between HNF-4 and Sp TFs contribute to the expression of several genes in cancer cells. HNF4 and Sp binding sites cooperatively activate the expression of the erythropoietin (Epo) gene in Hep3B cells [90]. The maximal expression of Epo is observed in hypoxic conditions and treatment with TGFβ, and this results in a complex containing not only HNF4 and Sp1 but also SMAD3, HIFIα:Arnt and CBP/p300 cofactors. HNF4/Sp1 interactions that induce eosinophil RNase2 in cancer cells [91] are also complex, where HNF-4/Sp1 and HNF-4/Sp3 activate the expression of human haem oxygenase-1 in hepatoma cells [92]. Zannis and coworkers have reported the induction of the apolipoprotein C-III (apoC-III) regulatory element in both cancer and non-cancer cell lines. In Caco2 colon cancer cells, HNF-4-Sp1 distal interactions contribute to apoC-III activation [93], and in other cells, there are additional cis-elements and factors, including EAR-2 and EAR-3 [94,95], which contribute to the expression of apoC-III.

Other sub-group II receptors: Among the remaining NRs in this subgroup, testicular receptor 4 (TR4) is the receptor that activates genes (e.g., apoE) in cancer cells, and this involves both TR4 response elements alone and in combination with Sp1 [96]. COUP-TF interacts directly with Sp1 and COUP-TF/Sp1 alone and, in combination with other nuclear factors, regulates the expression of NGF1-A [97] and human immunodeficiency virus type 1 (HIV-1) [98,99] in non-cancer cells. In addition, myocardial fatty acid β-oxidation genes are depressed by COUP-TF interactions with both Sp1 and Sp3 [100]; COUP-TF1 also interacts with Sp1 and Sp3 [101]; and COUP-TFII/Sp1 regulates antimullerian hormone receptor type 2 in mouse Leydig cells [102]. TLX/Sp1 interactions induce the pro-neuronal MASH-1 gene in neuronal cells [103]. Thus, with the exception of photoreceptor cell-specific nuclear receptor (PNR), sub-group II receptors interact with Sp1 or Sp3 to modulate gene transcription.

## 4. Subgroup I Receptors and Their Interactions with Sp TFs

The largest number of nuclear receptors are members of subgroup 1, and they act primarily as heterodimers, with RXR as the partnering receptor. Included in this group of 20 receptors are the thyroid hormone receptor (TRα/β), retinoic acid receptor (RARα/β/γ), PPARα/β/Ɣ (NR1C1-3), reverse etb (Rev-Erβα/β, NR1D1-2), retinoic acid-related orphan receptor (RORα/β/Ɣ, NR1F1-3), farnesoid X receptorα (FXRα, NR1H4), liver X receptor (LXR) and constitutive androgen receptor (CAR, NR1I3). Sp TFs form complexes with subgroup1 receptors, which also include other nuclear factors such as coactivators, corepressors and basal transcription factors (BTFs).

Peroxisome proliferator-activated receptor (PPAR): PPARƔ interacts with Sp1 (and Sp3) to regulate gene expression in both cancer and non-cancer cells, and most of these responses include interactions with other nuclear factors. There are also a few examples of PPARα/Sp-mediated responses, but not in a cancer cell context. PPARƔ primarily inhibits the growth of cancer cells; emodin decreases IGFBP1 [104]; cladosporol, rosiglilazone and a bis-indole PPARƔ ligand induce p21 [105,106,107]; and thiazolidinediones (TZDs) decrease an insulin receptor [108] via PPARƔ/Sp1 in cancer cell lines (Figure 5). Phorbol ester upregulated the expression of the resistin gene via PPARƔ/Sp1 in U937 cells [109]. The ligand-induced and PPARƔ/Sp1-dependent activation of p21 in pancreatic thyroid and colon cancer cells involved the same proximal (−124 to −60) GC-rich sites in the p21 gene promoter. PPARƔ/Sp1 interactions were also involved in decreased fatty acid synthesis [110]; induction of adipocyte triglyceride lipase (Atgl); induction of acetyl-CoA synthetase (AACS); and decreased follistatin, KDR, angiotensin type 1 receptor (AT1R) and the thromboxane receptor in non-cancer cell lines [110,111,112,113,114,115,116]. PPARα agonists also decreased the PPARα/Sp1-dependent expression of KDR and Cyp7b1 and induced pl6NK4a in non-cancer cell lines [117,118,119].

Retinoic acid receptors: RAR and other nuclear receptors were reported to bind Sp1 and induce transactivation, and these studies also showed that the two proteins could interact at both GC-rich and retinoid acid response elements (RARE), and the C-terminal of Sp1 is required [120]. Ligands activated RARα/Sp1-induced monoamine oxidase B (MAO-B) [121], CADM1 [122], E-cadherin [123] and the folate receptor-β gene (FRβ) [124] in cancer cells (Figure 5). In leukemia cells, retinoic acid activated RARα/Sp1, whereas RARβ and RARγ were associated with both AP1 and Sp1 binding sites in the FRβ gene promoter, and the ets protein was also associated with Sp1 [124]. In addition, the oncogenic promyelocytic leukemia zinc finger–retinoic acid receptor α (PLZF-RARα) fusion gene inhibited p21 expression through interaction with Sp1 bound to the proximal GC-rich sequences in the p21 gene promoter and also with an RARE. The repressive response induced by this oncogene is due to the recruitment of other nuclear cofactors, such as corepressors and deacetylases [125]. In non-cancer cells, RARα/Sp1 enhances induction pyruvate dehydrogenase kinase 4 (PDK4), guanylyl cyclase/atrial natriuretic peptide receptor-A (Npr1), apelin and TNFα-inducible protein 3-interacting protein 1 (TNIP1) [126,127,128,129]. Ets protein–Sp1 interactions are important for the induction of Npr1 [127], and Kruppel-like factor 5 also interacts with RARα in the upregulation of apelin in vascular smooth muscle cells [128].

Other sup-group I receptor: The vitamin D receptor (VDR, NR1I1) and its interactions with Sp TFs play a role in the activation of p27Kip1 [130,131] and CYP24A1 [132] in colon cancer cells and CD14 in U937 cells [133]. In the latter cell line, the response primarily involves Sp3. Thyroid hormone activates constructs associated with a thyroid hormone-responsive gene and the human Type 1 iodothyronine deiodinase gene (dio1) (Figure 6) where there are cooperative functional interactions between Sp1 and TR where the Sp1 binding are proximal and the TREs are distal. The location of the distal TRE site from the Sp1 element is different, and in the dio1 gene promoter, there is also a functional proximal TRE site [134,135]. Zearalenone enhances PXR-Sp1 interaction to decrease endothelial NO synthase (eNOS) in endothelial cells [136], and LXR-Sp1 (separated elements) interactions modulate the expression of the ATP-binding cassette transporter A1 (ABCA1) [137]. In addition, the ligand-binding domain Rev-ERBα interacts with Sp1 bound to the connexin43 gene promoter to activate gene expression in HEK293T cells [138].

## 5. Subgroup V, VI and Atypical Receptors Interactions with Sp TFs

These sub-groups contain a relatively small number of NRs, and these include steroidogenic factor-1 (SF1, NR5A1), liver receptor homolog-1 (LRH-1, NR5A2), and germ cell nuclear factor (GCNF). Both DAX1 and SHP are atypical NRs that lack a DNA-binding domain. Among these NRs that act through interactions with Sp1, SF-1 is the major contributor.

SF-1: Most SF-1 interactions with Sp1 are associated with induction/repression of gene expression. However, one study reported that Sp1 and SF-1 interact through the N-terminal regions of NF-1 and the DBD of Sp1, inducing CYP11A in Y1 adrenal tumor cells and bovine luteal cells [139]. This interaction involved cis-elements for both Sp1 and NF-1 (i.e., NF-1-Sp1), and a similar mechanism was reported for NF-1-Sp1-mediated activation of the human steroidogenic acute regulatory (STAR) gene in Y-1 cells [140]. Two studies reported that the induction of rat luteinizing hormone-β (LHβ) was dependent on both NF-1 and Sp1 located in proximal and distal positions in the LHβ promoter [141,142] (Figure 7). The proposed mechanism also involved the nuclear cofactor SNURF, which interacts with both NF-1-Sp1 interactions and may be indirect through an association with SNURF (Figure 7) [142]. Another example of a complex nuclear receptor-Sp1-mediated response is the regulation of the mouse aldose-reductase-like gene (MVDP gene; AKR1B7 protein) in Y1 cells. The gene promoter contains four key cis-elements that bind SF-1, NF-1, C/EBP and Sp1. The results of deletion and mutational analyses showed that the SF-1 and NF-1 sites were required for basal and forskolin-induced activity; however, the fold-induction was unchanged. Both NF-1 and Sp1 are required for basal expression of MVDP, while Sp1 and C/EBPs interact and mediate forskolin inducibility (Figure 6) [143]. SF-1-Sp also regulates the expression of the human SRY gene, which is suppressed by cAMP-dependent phosphorylation of NF1 [144], and SF-1-Sp1 also regulates the expression of the Oct4 gene, which is downregulated by retinoic acid in stem cells [145].

Other receptors: Sp2 and KLF-6 bind matrix metalloproteinase-9 (MMP-9) gene promoter, and this results in the silencing of this gene and activation of FXR, whereas SHP interacts with Sp2 and KLF6 to upregulate the expression of MMP-9 [146]. LRH-1 and Sp1 and Prospero-related homeobox protein (Prox1) play a role in hepatitis B virus gene transcription; however, the interaction of LRH-1 (NR5A2) and Sp1 is not directly addressed [147].

## 6. Subgroup IV Receptors and Their Interactions with Sp1

Subgroup IV contains members of the orphan NR subfamily of NR4A receptors, which include NR4A1 (Nur77, TR3), NR4A2 (Nurr1) and NR4A3 (Nor1). These receptors are immediate early genes induced by stressors and play an increasingly important role in maintaining cellular homeostasis and in pathophysiology [148,149]. Interactions of NR4A2 and NR4A3 with Sp1/Sp4 to activate or deactivate gene expression have not been characterized; however, there is considerable evidence that NR4A1 interacts with Sp1 and Sp4 to regulate several genes in cancer cell lines [150].

NR4A1 (Nur77): Studies on the identification function of NR4A1-regulated genes in cancer cells have been investigated in NR4A1 knockdown studies, and it was observed that several DEGs contained GC-rich promoter, and many did not contain cognate NBREs or NuREs. Survivin was first characterized as an NR4A1/Sp1-regulated gene in pancreatic cancer cells [151] followed by characterization of β1-integrin as an NR4A1/Sp1 regulated gene in breast [152], pancreatic and colon [153] cancer cells (Figure 8). In contrast, NR4A1/Sp4 regulated β1-integrin and the oncogenic PAX3-FOX01 fusion gene expression in rhabdomyosarcoma cells [154], whereas NR4A1/Sp1 regulated G9a expression in the same cell lines [155]. Other studies in cancer cell lines showed that several NR4A1-regulated integrins, including β1-, β4-, α5- and α6-integrins, were coregulated by NR4A1 interactions with Sp1, Sp3 and Sp4; however, the role of individual Sp TFs was gene and cell context-dependent [152]. PD-L1 was also an NR4A1/Sp1-regulated gene in breast cancer cells [156]. Recent studies show that HuR and IDH-1 were regulated by NR4A2/Sp4 in pancreatic cancer cells [157].

## 7. Nuclear Receptor/Sp Variability 

The nuclear receptor/Sp complex exhibits flexibility in the regulation of genes based on the structure of the receptor, the ligand and variability of the Sp TF (i.e., Sp1, Sp3 and Sp4), and this is illustrated in Figure 2 and Figure 3, where differences in ER (α and β) and Sp (Sp1 and Sp3) enhance or inhibit gene expression. Figure 9 illustrates the variability in the regulation of vascular endothelial growth factor receptor2 (KDR) expression. In ER-positive ZR-75 breast cancer cells, treatment with E2 results in the induction of KDR, and this is associated with ERα/Sp3 and ERα/Sp4 but not ERα/Sp1 binding the proximal Sp binding sites in the KDR promoter [156]. In contrast, E2 downregulated KDR expression in ER-positive MCF-7 breast cancer cells, and this involved ERα/Sp1 and ERα/Sp3 but not ERα/Sp4, and this is accompanied by the recruitment of corepressors SMRT and NCoR [38]. In both studies, the two proximal −60 and −37 Sp binding sites were primarily responsible for these responses in breast cancer cells. PPARƔ ligands, such as pioglitazone and prostaglandin J2, enhance PPARƔ/Sp1, but not PPARƔ/Sp3 binding, in a gel shift assay; PPARƔ/Sp1 activates KDR expression by binding the proximal GC-rich region of the KDR promoter in retinal capillary endothelial cells [114]. Similar results were observed in HUVEC cells where ligand-activated PPARα/Sp1 decreased KDR expression, and this was associated with the −60 and −37 Sp binding sites [117]. Thus, NR/Sp induced or repressed the expression of KDR, and this response was nuclear receptor-, Sp1-, Sp3- and cell context-dependent.

The cyclin-dependent kinase inhibitor p21 promoter also contains proximal GC-rich Sp binding sites, which are targets for nuclear receptor/Sp interactions, and many studies show that functional responses are dependent on the Sp1–6 clusters in the p21 promoter. Ligands for PPARƔ, AR, PR and RARα activate p21 expression in cancer cells [72,106,107,158] through interactions of NR/Sp1 or NR/Sp4 with two or more members of the Sp1–6 binding sites in the proximal region of the p21 gene promoters (Figure 10). In addition, there is evidence for AR-dependent induction to include interactions of AR with a distal androgen response element [72]. The fusion PLZ-RARα acts as a unique transcriptional repressor of p21 not only through interactions with the proximal Sp binding sites but also with a retinoic acid response element (RARE) and a distal p53 responsive element [124]. In contrast, several NRs modulate p21 expression by interacting with cognate NR response elements; however, the possibility of a role for Sp TFs could not be excluded for some of these studies. For example, the overexpression of HNF4α induced p21 in prostate cancer cells, and the results of several experiments, including ChIP assays, suggest that this response is due to HNF4α interactions with distal (−980 to −755) response elements [159]. In contrast, RARα and the androgen receptor regulate p21 expression in colon and breast cancer cells, respectively, through Sp-independent promoter pathways associated with binding their cognate response elements in the p21 promoter [160,161]. Thus, several hormone receptors activate p21 in cancer cells, and these responses can be Sp-dependent or independent.

## 8. Conclusions

In summary, the NR/Sp complex modulates the expression of diverse genes in cancer and non-cancer cell lines, and this involves binding to GC-rich gene promoter sites. The induction or repression of gene expression by NR/Sp is dependent on the receptor, Sp TFs (Sp1, Sp3 or Sp4), specific GC-rich sequences, ligand structure and cell context. Moreover, other nuclear cofactors are also critical elements in functional NR/Sp-mediated gene expression, and these have been identified only in a limited number of studies. Unfortunately, the available data is not sufficient to predict genes that can be regulated by NR/Sp1. Many Sp- and Sp-regulated genes in cancer cells play a role in pro-oncogenic pathways that can exacerbate or protect against or enhance cell proliferation, survival and migration/invasion. For example, the activation of cyclin-dependent kinase inhibitors p21 and p27 or the inactivation of PD-L1 or survivin via NR/Sp would inhibit carcinogenesis, whereas activation of these genes would have the opposite effect. These responses can be modified not only by NR agonists or inverse agonists but also by a large number of drugs and inducers of reactive oxygen species (ROS) that activate pathways that downregulate Sp1, Sp3 and Sp4 [162]. Thus, it is important to further characterize NR/Sp-regulated pro-oncogenic genes since they can be targeted by both NR ligands and drugs that downregulate Sp TFs [162].

## Figures and Tables

**Figure 1 cancers-17-00284-f001:**
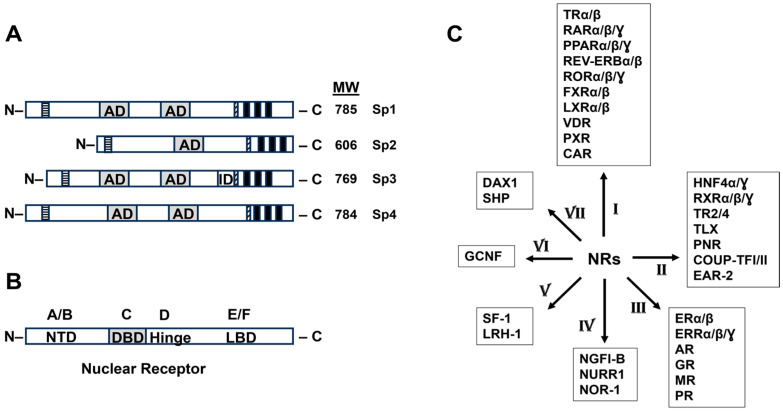
Structural domains and molecular weights (MW) of Sp1, Sp2, Sp3 and Sp4 (**A**) and NRs (**B**), classification of NRs (**C**) based on structural homology. AD = activation domain; ID = inhibitory domain; NTD = N-terminal domain ((**B**). site A/B); DBD = DNA binding domain ((**B**). site C); Hinge ((**B**). site D); LBD = ligand binding domain ((**B**). site E/F).

**Figure 2 cancers-17-00284-f002:**
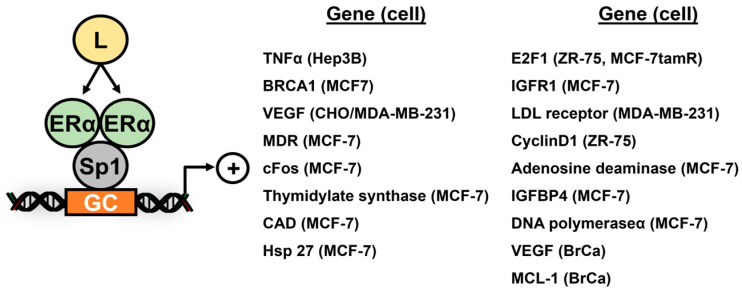
Hormonally active genes regulated by ERα/Sp1, where L is the estrogenic ligand [18,19,20,21,22,23,24,25,26,27,28,29,30,31,32,33,34,35,36,37]. ER/Sp1 signifies that ER directly binds Sp1 but not promoter DNA.

**Figure 3 cancers-17-00284-f003:**
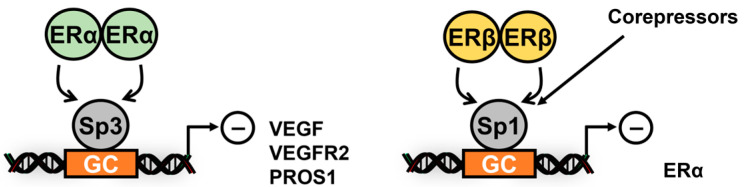
R-Sp interactions that involve ERα, ERβ, Sp1 and Sp3 and repress expression of VEGFR2 [38], PROS1 [39] and ERα [40].

**Figure 4 cancers-17-00284-f004:**
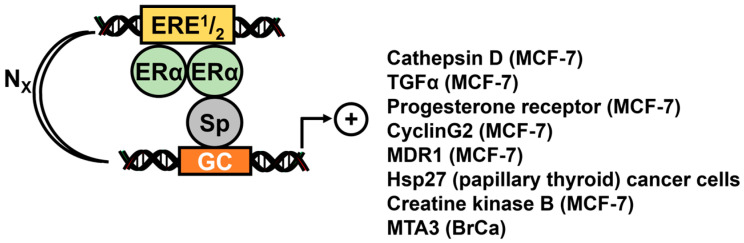
Hormonal activation of genes in cancer cells through ER-Sp bound to GC-rich(N)XERE^1^/_2_ [50,51,52,53,54,55,56,57]. ER-Sp1 indicates that ER binds Sp1 and also promotes DNA (e.g., ERE or an ERE^1^/_2_).

**Figure 5 cancers-17-00284-f005:**
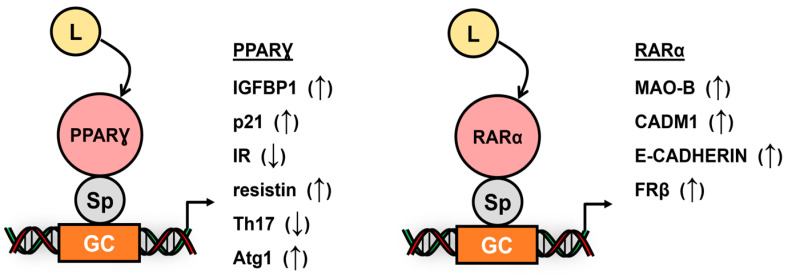
Ligand-dependent induction (↑) or repression (↓) of PPARƔ- and RARα-dependent genes through nuclear receptor/Sp interaction with Sp bind sites [48,109,120,121,122,123,124,125].

**Figure 6 cancers-17-00284-f006:**
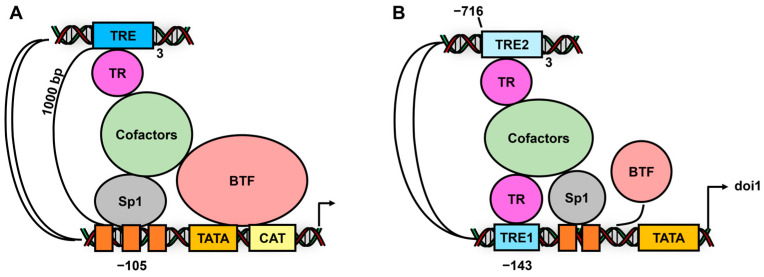
TR-Sp1 interactions. (**A**) Thyroid hormone interacts with 3 tandem thyroid hormone receptor response elements and induces TR-Sp1 CAT activity, and this involves a 1000 bp distance between the dependent Sp1 and (TR) binding sites [134]. (**B**) Activation of the human type1 iodothyronine deiodinase gene (dio1) [135] involves Sp1 and distal/proximal.

**Figure 7 cancers-17-00284-f007:**
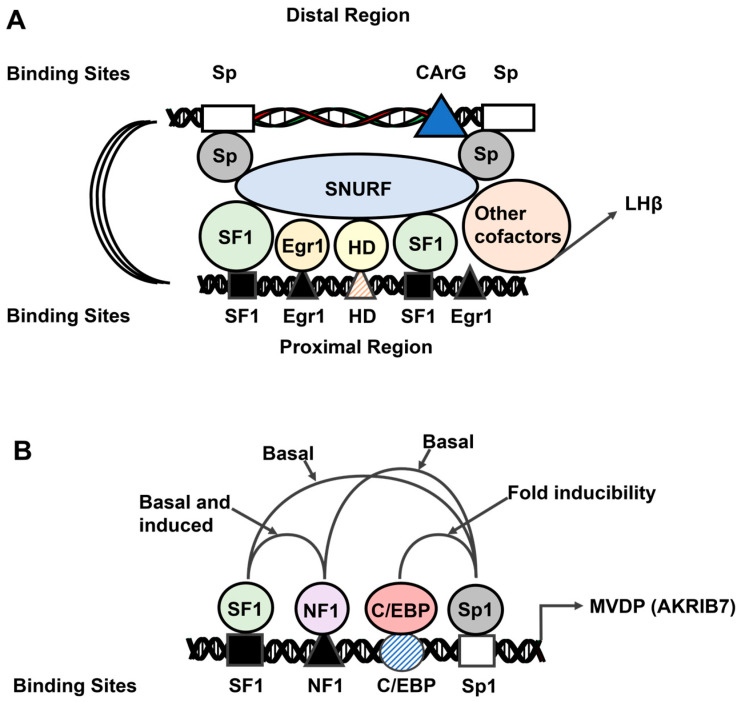
Models illustrating the role of SF-1 and Sp1 in modulating expression of LHβ (**A**) and MVDP (**B**) and the importance of other nuclear cofactors and DNA-bound transcription factors [141,142].

**Figure 8 cancers-17-00284-f008:**
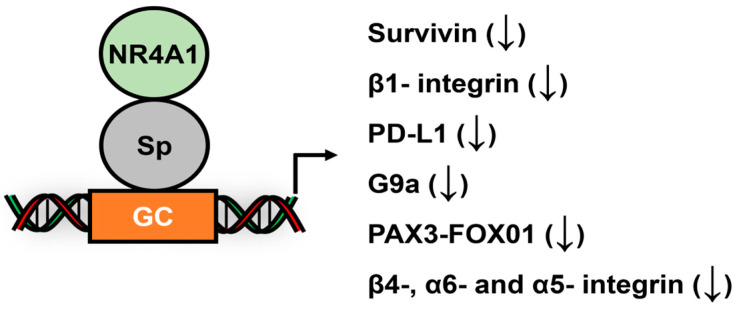
NR4A1/Sp1 interactions at Sp promoter binding sites inhibit expression of multiple genes in cancer cells [150]. The (↓) indicated ligand-induced gene repression.

**Figure 9 cancers-17-00284-f009:**
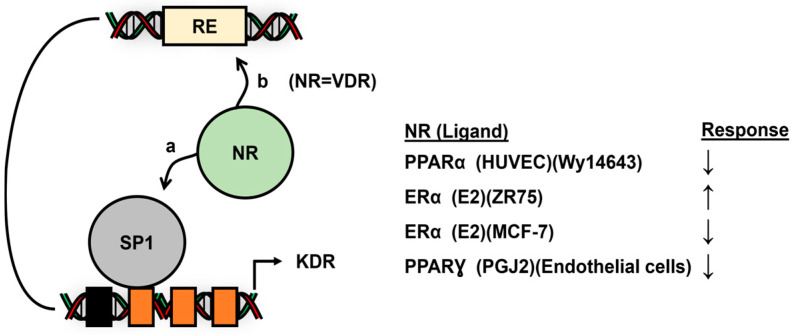
Modulation of KDR expression by various NRs and their ligands, which involves interactions of the NR with Sp1 (**a**) and also (**b**) distal response elements (RE) [38,114,117,156]. (↓) and (↑) denote ligand-dependent repression and induction of KDR, respectively.

**Figure 10 cancers-17-00284-f010:**
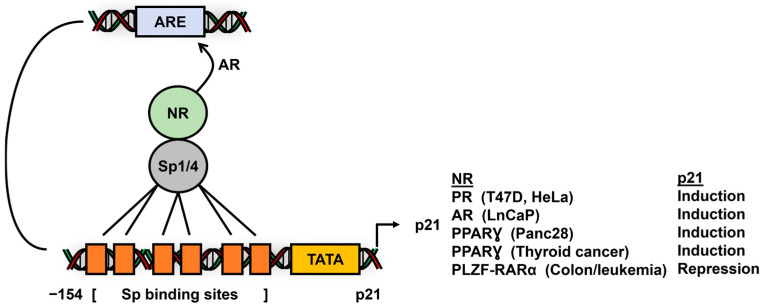
NR/Sp1-mediated induction of inhibition of p21 gene expression in cancer cells.

## Data Availability

Not applicable.

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
