# Peer review of "Activation of Genes by Nuclear Receptor/Specificity Protein (Sp) Interactions in Cancer"

_cancers, 2025, doi:10.3390/cancers17020284_

Round 1

Reviewer 1 Report

Comments and Suggestions for Authors

I appreciate the hard work the team invested in this complex review, which approaches a topic of high importance nowadays. The manuscript is well written, with attention to details and well-structured. The subparagraphs were well-chosen and the content appears explanatory enough, with some data being illustrated in the 10 figures. I carefully read the iThenticate report and it looks fine.

I have a few comments, listed for consideration, below:

a.      Major:

1.      Please emphasize more on what is the practical role of your review. What are the possible implications?

2.      Please elaborate proper directions for further research.

3.      Please summarize data that appear in the main text (regarding existing studies) in pertinent tables, much easier to be followed and retained.

4.      Please elaborate appropriate Conclusion.

b.      Minor:

1.      The “Simple summary” mentions the aim of the review, but this aim should also be included in the Abstract.

2.      Lines 41-42 (at the beginning) mention “this review is focused on the transcriptional activities of nuclear receptors (NR)/Sp complexes in cancer.”. Please delete this from here, as it appears in the aim, by the end of Introduction. The place of the aim is by the end of Introduction, correctly inserted.

3.      Figure 1 legend. A: “Structural domains of Sp1, Sp3 and Sp4 (A)” – Please add Sp 2, as it appears in the figure.

4.      In Figure 1 C, please write VII in the Figure, for DAX1 and SHP.

5.      Figure 9 legend: Please detail a and b.

Thank you

Author Response

a. Major

Comments 1: Please emphasize more on what is the practical role of your review. What are the possible implications?

Response 1: The review now further emphasizes some implication of the NR/Sp pathway (lines 25-26, 86-95).

Comments 2: Please elaborate proper directions for further research.

Response 2: This additional text (lines 86, 95, 433-435) emphasizes some directions for future research (also see the final section VII).

Comments 3: Please summarize data that appear in the main text (regarding existing studies) in pertinent tables, much easier to be followed and retained.

Response 3: We have considered using Tables to summarize the data in the text (suggested by Reviewer 1) however, we have included 10 data-rich Figures and addition of Tables would be redundant.

Comments 4: Please elaborate appropriate Conclusion.

Response 4: The conclusions are included in #1 and #2 above and these have now been re-emphasized at the end of the review (lines 433-435).

b. Minor

Comments 1: The “Simple summary” mentions the aim of the review, but this aim should also be included in the Abstract.

Response 1: The aim (in the Summary) is now incorporated into the Abstract (lines 25-26).

Comments 2: Lines 41-42 (at the beginning) mention “this review is focused on the transcriptional activities of nuclear receptors (NR)/Sp complexes in cancer.”. Please delete this from here, as it appears in the aim, by the end of Introduction. The place of the aim is by the end of Introduction, correctly inserted.

Response 2: Lines 41-42 have now been deleted.

Comments 3: Figure 1 legend. A: “Structural domains of Sp1, Sp3 and Sp4 (A)” – Please add Sp 2, as it appears in the figure.

Response 3: Sp2 has been added.

Comments 4: In Figure 1 C, please write VII in the Figure, for DAX1 and SHP.

Response 4: Figure 1 (add VII) has been changed.

Comments 5: Figure 9 legend: Please detail a and b.

Response 5: The legend for Figure 9 has been changed.

Reviewer 2 Report

Comments and Suggestions for Authors

Tracking #: Cancers – 3385590Stephen safe et al.

The review article entitled “Activation of genes by Nuclear Recetor/Sp interactions in cancer” provides an update on the genes jointly regulated by proteins from the two families formed by nuclear receptors and Sp proteins in the context of different types of cancer.

Given the importance of these signalling pathways and the therapeutic potential of these proteins, this review is interesting and original, and adequately documented with an extensive literature. It represents a major effort.

On the whole, the text is clear. The abstract clearly announces what the reader will find in the paper. The figures (except for figure 5) are correct and sufficiently informative to illustrate the text.

The following is a list of few points that should be clarified and/or improved:

1/ Although the bibliography is complete, most of the references are old. This may reflect the fact that less research has been carried out on the subject in recent years, but it doesn't detract from the interest of this review, which takes stock of the situation and may stimulate new research. The need for further studies to clarify NR/Sp-controlled regulation is also apparent from the authors' summary at the end of the text.

2/ For nuclear receptor ligands, the authors refer to agonists and inverse agonists. It would be beneficial for the reader if these classes of ligands were defined in the context of nuclear receptors.

3/ Line 105 and Figure 3: Can authors indicate which corepressors are involved in this example of Sp and ER?

4/ Line 199: Surprisingly, the DNA binding site in the example shown is a DR4 sequence for the RXR-RAR heterodimer. These heterodimers generally interact preferentially with other sequences (DR5, DR2, DR1, ...). Can the authors comment on this fact?

5/ Line 191: Although RXRs occupy a central position in NR signaling, their endogenous ligands (rexinoids) and the physiologic functions of such ligands have not been yet conclusively determined. Initial studies led to the discovery of 9-cis-retinoic acid (9cisRA) as a high-affinity ligand for all three RXRs. However the existence of physiological rexinoids, their characterization and elucidation of their roles in biological systems remain highly controversial, and an unsolved question. The authors should not assert that 9cisRA is a bona fide ligand for RXR.

6/ The case illustrated in Figure 5 for PPAR and RAR is unclear to my mind and deserves a more detailed description. In particular, how Sp1 is involved.

7/ Authors use a trivial name for nuclear receptors (for instance ER for estrogen receptors). A logical numbering system and receptor code, supporting the trivial names, was made by the International Committee of Pharmacology Committee on Receptor Nomenclature and Classification (NC-IUPHAR). In each manuscript dealing with nuclear receptors, it is recommended that the receptors be identified by the official names at least once in the summary and the introduction. Once the name has been established, authors may use the trivial name for the remainder of the manuscript.  For instance, the trivial names and the formal nomenclature for ER and LXRα are NR3A and NR1H3, respectively. Please the modifications must be made at the first mention of each receptor

8/ Mistake: line 234.  The abbreviation PXR specified for liver X receptor, which is LXR. PXR is the Pregnane X receptor (NR1I2).

Author Response

Comments 1: Although the bibliography is complete, most of the references are old. [...] The need for further studies to clarify NR/Sp-controlled regulation is also apparent from the authors' summary at the end of the text.

Response 1: I agree with the Reviewer that many of the references are older, and this may be due to the lack of appreciation for the role of nuclear receptors as cofactors of Sp TFs. I’m hoping that this comprehensive review will alert researchers to the fact that this mechanism may be more important and widespread for many Sp-regulated genes.

Comments 2: For nuclear receptor ligands, the authors refer to agonists and inverse agonists. It would be beneficial for the reader if these classes of ligands were defined in the context of nuclear receptors.

Response 2: The definitions of NR agonists, antagonists and inverse agonists have been included at the end of section I (lines 85-90).

Comments 3: Line 105 and Figure 3: Can authors indicate which corepressors are involved in this example of Sp and ER?

Response 3: The corepressors include RIP140 and the NCoR-SMRT-HDAC3 complex; this is now indicated in the text (line 119-120)

Comments 4: Line 199: Surprisingly, the DNA binding site in the example shown is a DR4 sequence for the RXR-RAR heterodimer. These heterodimers generally interact preferentially with other sequences (DR5, DR2, DR1, ...). Can the authors comment on this fact?

Response 4: The interactions with DR4 site were confirmed in mutation experiments and the surprising interactions at this site (-1530 to -1516) might be due to concurrent interactions with Sp1 bound more proximal GC-rich sites at -145 tp -121 in the thrombomodulin promoter.

Comments 5:  Line 191: Although RXRs occupy a central position in NR signaling, their endogenous ligands (rexinoids) and the physiologic functions of such ligands have not been yet conclusively determined. [...] The authors should not assert that 9cisRA is a bona fide ligand for RXR

Response 5: We have corrected the assertion on line 191 indicating that 9-cis-retinoic acid is not a bona fide endogenous ligand for RXR.

Comments 6: the case illustrated in Figure 5 for PPAR and RAR is unclear to my mind and deserves a more detailed description. In particular, how Sp1 is involved.

Response 6: Figure 5 is confusing and has now been completely revised

Comments 7: Authors use a trivial name for nuclear receptors (for instance ER for estrogen receptors). [...] Please the modifications must be made at the first mention of each receptor.

Response 7: The trivial and formal names for each receptor are now indicated when they are initially mentioned in the text.

Comments 8: Mistake: line 234.  The abbreviation PXR specified for liver X receptor, which is LXR. PXR is the Pregnane X receptor (NR1I2).

Response 8: The mistake on line 234 has been corrected.

Reviewer 3 Report

Comments and Suggestions for Authors

Comments on the manuscript cancers-3385590 this is a comprehensive review on the role played by nuclear receptors, and specificity protein transcription factors in the regulatory pathways of carcinogenesis. In general, the background and conclusions are appropriate for the topic, the figures are clear and easy to interpret. Just one suggestion: specify in the caption of figure 1A what is meant by 785, 606…

Author Response

Comments 1: Just one suggestion: specify in the caption of figure 1A what is meant by 785, 606…

Response 1: This definition is the molecular weights (MW) and has been designated in Figure 1 and its caption.

Reviewer 4 Report

Comments and Suggestions for Authors

cancers-3385590

The manuscript cancers-3385590 entitled Activation of Genes by Nuclear Receptor/Sp Interactions in Cancer by Stephen Safe and coworkers, is a review about the human nuclear receptor (NR) superfamily that are ligand-activated transcription factors that play a key role in maintaining cellular homeostasis and in pathophysiology. NRs are important drug targets for both cancer and non-cancer endpoints as ligands for these receptors can act as agonists, antagonists or inverse agonists to modulate gene expression.

Several studies showed that a number of NR-regulated genes did not directly bind their corresponding cis-elements and promoter analysis identified that NR-responsive gene promoters contained GC-rich sequences that bind specificity protein 1 (Sp1), Sp3 and Sp4 transcription factors (TFs). Subsequent studies showed that many NRs directly bind Sp1 (or Sp3 and Sp4) and that the NR/Sp complexes bind GC-rich sites to regulate gene expression and the NR acts as a ligand-modulated nuclear cofactor. In addition, several reports show that NR-responsive genes contain cis elements that bind both Sp TFs and NRs, and mutation of either cis-element results in loss of NR-responsive (inducible and/or basal). These genes involve interactions between DNA bound Sp TFs with proximal or distal DNA-bound NRs and in some cases other nuclear cofactors are required for gene expression.

This review work is well written and updated.

The narrative quality is very high

Figures are informative.

Language is good.

Conclusion consistent with the text

References appropriated.

The text should be better organised.

Line 36: there is an empty space

Line 88, 187, 227, 294, 329, 353: the title should be made more clear. For example using a letter like a), b), c)…. or number (1, 2, 3) to the subsections

Comments on the Quality of English Language

engish is good

minor revisions

Author Response

Comments 1: Line 36: there is an empty space

Response 1: Line 36: the empty space separates the Key Words from the Introduction section.

Round 2

Reviewer 1 Report

Comments and Suggestions for Authors

I agree with this new version of the manuscript and I support its publication, in the present form.